# Safety and Feasibility of Neoadjuvant-Modified FOLFIRINOX in Elderly Patients with Pancreatic Cancer

**DOI:** 10.3390/cancers16142522

**Published:** 2024-07-12

**Authors:** Yoshitaro Shindo, Tatsuya Ioka, Yukio Tokumitsu, Hiroto Matsui, Masao Nakajima, Yuta Kimura, Yusaku Watanabe, Shinobu Tomochika, Yuki Nakagami, Ryouichi Tsunedomi, Michihisa Iida, Hidenori Takahashi, Hiroaki Nagano

**Affiliations:** 1Department of Gastroenterological, Breast and Endocrine Surgery, Yamaguchi University Graduate School of Medicine, Ube 755-8505, Yamaguchi, Japan; y.shindo@yamaguchi-u.ac.jp (Y.S.); yt790604@yamaguchi-u.ac.jp (Y.T.); matsui-h@yamaguchi-u.ac.jp (H.M.); masao-y@yamaguchi-u.ac.jp (M.N.); ykimura@yamaguchi-u.ac.jp (Y.K.); yusakuw@yamaguchi-u.ac.jp (Y.W.); bu-shino@yamaguchi-u.ac.jp (S.T.); tsune-r@yamaguchi-u.ac.jp (R.T.); miida@yamaguchi-u.ac.jp (M.I.); hdnrtkhs@yamaguchi-u.ac.jp (H.T.); 2Department of Oncology Center, Yamaguchi University Hospital, Ube 755-8505, Yamaguchi, Japan; ioka64t@yamaguchi-u.ac.jp; 3Department of Data Science, Shimonoseki City University, Shimonoseki 751-8510, Yamaguchi, Japan; nakagami-yu@shimonoseki-cu.ac.jp

**Keywords:** resectable pancreatic cancer, borderline resectable pancreatic cancer, elderly patients, neoadjuvant chemotherapy, FOLFIRINOX

## Abstract

**Simple Summary:**

Neoadjuvant chemotherapy (NAC) is effective in improving the prognosis of patients with resectable and borderline resectable pancreatic cancer (PC). A modified FOLFIRINOX (mFOLFIRINOX) regimen is widely used as the standard therapy. However, owing to its toxicity, it is not applied in patients older than 75 years. Furthermore, elderly patients often have concomitant conditions and are at high risk for adverse effects associated with chemotherapy. Given the rapid increase in the number of elderly patients, it is necessary to determine whether NAC can be performed in both non-elderly and elderly patients. Thus, in this study, we investigated the safety and feasibility of neoadjuvant mFOLFIRINOX in elderly patients with PC. We found that the perioperative and postoperative outcomes of elderly patients who received neoadjuvant mFOLFIRINOX were comparable to those of non-elderly patients.

**Abstract:**

The optimal treatment strategy for neoadjuvant chemotherapy in elderly patients with pancreatic cancer (PC) remains unclear. Hence, this study was aimed at evaluating the safety and feasibility of neoadjuvant-modified FOLFIRINOX (mFOLFIRINOX) in elderly patients with PC. We retrospectively collected data from 62 patients who received neoadjuvant mFOLFIRINOX between May 2015 and October 2023 and comparatively analyzed the clinicopathological data and outcomes between the non-elderly group (age: <75 years) and elderly group (age: >75 years). The non-elderly and elderly groups comprised 39 and 23 patients, respectively. Although elevated levels of aspartate aminotransferase (*p* = 0.0173) and alanine aminotransferase (*p* = 0.0378) and nausea (*p* = 0.0177) were more frequent in the elderly group, the incidence of severe adverse events was similar between the groups. Intergroup differences in resection rate (*p* = 0.3381), postoperative severe complication rates (*p* = 0.2450), and postoperative hospital stay (*p* = 0.3496) were not significant. Furthermore, no significant intergroup differences were found in survival in either the whole or the resection cohorts. The perioperative and postoperative outcomes of elderly patients treated with neoadjuvant mFOLFIRINOX were comparable with those of non-elderly patients. Neoadjuvant mFOLFIRINOX should be considered a feasible option for elderly patients with PC.

## 1. Introduction

Pancreatic cancer (PC), one of the most lethal cancers, is the seventh leading cause of cancer-related deaths worldwide and the fourteenth most common cancer [1]. By 2030, PC is predicted to become the second leading cause of cancer mortality in Western countries [2]. Although radical resection is the only curative treatment for PC, most patients are diagnosed with unresectable or metastatic disease, with resectable tumors reported in only 20% of the cases at diagnosis [3]. Despite curative resection, most patients develop postoperative recurrence, and the prognosis, including that for unresectable and metastatic stages, remains unsatisfactory, with the 5-year overall survival rate being 6% [4].

Neoadjuvant chemotherapy (NAC) is an effective strategy for improving R0 resection and prognosis in patients with resectable (R) and borderline resectable (BR) PC. The Prep-02/JSAP-05 study showed that NAC with gemcitabine plus S-1 (GS) afforded a significant survival advantage over upfront surgery for patients with R and BR PC [5]. After fluorouracil, leucovorin, irinotecan, and oxaliplatin (FOLFIRINOX) and gemcitabine with nab-paclitaxel (GnP) were shown to improve the prognosis of patients with advanced PC, regimens such as NAC were introduced [6,7,8,9]. Among these regimens, FOLFIRINOX is associated with a high incidence of adverse events (AEs) and therefore has limited indications. In Japan, the modified FOLFIRINOX (mFOLFIRINOX) regimen, in which rapid intravenous 5-FU (bolus 5-FU) is omitted and the irinotecan dose is reduced to 150 mg/m^2^ to decrease the toxicity of FOLFIRINOX therapy, is widely used as the standard therapy. However, it is more toxic than GnP and is not administered to patients older than 75 years of age [10]. Currently, there are no prospective randomized trials to evaluate the efficacy and toxicity of mFOLFIRINOX in patients older than 75 years. Compared with non-elderly patients, elderly patients often have concomitant conditions, including heart disease, hypertension, and diabetes, and are considered to have higher risks associated with chemotherapy. Because the number of elderly patients has been increasing rapidly in recent years, it remains to be determined whether NAC can be performed in both non-elderly and elderly patients. Surgical opportunities should not be missed due to associated AEs during NAC in patients with R and BR PC. Hence, this study was aimed at evaluating the safety and feasibility of neoadjuvant mFOLFIRINOX in elderly patients with PC.

## 2. Materials and Methods

### 2.1. Study Design and Patients

This single-center retrospective cohort study included patients who received neoadjuvant mFOLFIRINOX for R and BR PC between May 2017 and October 2023 at Yamaguchi University Hospital, Ube, Japan. All patients had histologically or cytologically confirmed PC, and NAC was initiated. Pretreatment resectability was evaluated using triple-phase helical computed tomography (CT), gadolinium ethoxybenzyl diethylenetriamine pentaacetic acid-enhanced magnetic resonance imaging, and fluorine-18 fluorodeoxyglucose-positron emission tomography/CT. The study was conducted in accordance with the Declaration of Helsinki and the protocol was approved by the institutional review board of Yamaguchi University Hospital (H2024-026). Informed consent was obtained from all patients.

### 2.2. NAC

The mFOLFIRINOX regimen consisted of oxaliplatin 85 mg/m^2^, irinotecan 150 mg/m^2^, leucovorin 200 mg/m^2^, and fluorouracil 2400 mg/m^2^. Patients were administered the treatment every 2 weeks. Patients with R, and BR PC received 6, and 6–12 cycles of treatment, respectively. Depending on AEs or the patient’s general condition, the dose was reduced or treatment was withdrawn appropriately by the attending physician. Relative dose intensity was calculated as the ratio of the actual dose administered to the standard dose during treatment.

### 2.3. Data Collection

Pretreatment data included age, sex, American Society of Anesthesiologists Physical Status, body mass index, Charlson comorbidity index (CCI), G8 Geriatric Assessment Tool scores (Appendix B), tumor location, tumor size, resectability classification according to the National Comprehensive Cancer Network criteria, carbohydrate antigen 19-9 level, maximum standardized uptake value, and uridine diphosphate glucuronosyltransferase 1A1 level. The G8 score ranges from 0 (heavily impaired) to 17 (not at all impaired). A G8 score ≤14 is defined as impaired [11,12]. Toxicity was graded using the National Cancer Institute Common Terminology Criteria for AEs version 5.0. Perioperative and postoperative data included operative procedure, operative time, blood loss, vascular resection, margin status, morbidity, mortality, and length of hospital stay. Postoperative pancreatic fistula (POPF) and delayed gastric emptying were defined according to the classification system of the International Study Group of Pancreatic Surgery [13,14]. Postoperative complications were defined according to the Clavien–Dindo (C-D) classification system [15]. PC staging was performed according to the eighth edition of the Union for International Cancer Control TNM Classification.

### 2.4. Statistical Analysis

Quantitative data are expressed as medians (ranges), and qualitative data as percentages. Categorical variables were compared using the chi-squared and Fisher’s exact tests. The Mann–Whitney U test was used to assess differences between the study groups. For the intention-to-treat analysis, overall survival (OS) was measured from the start date of NAC until death from any cause. Progression-free survival (PFS) was defined as the time from the start date of NAC to tumor progression or death from any cause. Regarding the analysis within the resection cohort, OS was defined as the time from the date of curative surgery until death from any cause. Recurrence-free survival (RFS) was defined as the time from the date of curative surgery to the date of first recurrence or death. Survival curves were analyzed using the Kaplan–Meier method, the log-rank test, and the Cox proportional hazards regression model. Age and other variables were included in the Cox model, stratified by average relative dose intensity (ARDI) and employing the stepwise Akaike Information Criterion (AIC) for variable selection when a *p*-value < 0.20 was observed in univariate analyses. Statistical analyses were conducted using GraphPad Prism V8.0 (GraphPad Software Inc., San Diego, CA, USA), and the R language (R Core Team URL https://www.R-project.org/ (accessed on 25 June 2024), Vienna, Austria). Statistical significance was set at *p* < 0.05.

## 3. Results

### 3.1. Patient Characteristics

Of the 62 consecutive patients who underwent neoadjuvant mFOLFIRINOX, 23 were aged ≥75 years (elderly), and 39 were aged <75 years (non-elderly). The baseline characteristics of the patients are summarized in Table 1. There were no intergroup significant differences in sex, American Society of Anesthesiologists Physical Status, body mass index, CCI, G8, tumor location, tumor size, resectability status, carbohydrate antigen 19-9 level, maximum standardized uptake value, and uridine diphosphate glucuronosyltransferase 1A1 level.

### 3.2. NAC

Table 2 summarizes NAC outcomes. The rates of NAC completion, dose reduction, and surgical resection did not differ between groups. However, the average relative dose intensity tended to be lower in the elderly group than in the non-elderly group, although the intergroup difference was not statistically significant (*p* = 0.0665). The major reason for not undergoing surgical resection in both groups was disease progression (*p* = 0.7008). Following NAC, one patient in the elderly group did not undergo surgery due to local progression. Four patients in the non-elderly group were not surgical candidates due to disease progression (liver metastasis in two, local progression in one, and peritoneal dissemination in one patient). During surgery, one patient in the elderly group was diagnosed with liver metastasis. Three patients in the non-elderly group were diagnosed with peritoneal dissemination.

### 3.3. AEs

Table 3 shows the AEs associated with NAC. Neutropenia, anemia, thrombocytopenia, elevated liver enzyme levels, fatigue, nausea, diarrhea, and sensory neuropathy were the most common AEs in both groups. Elevated aspartate aminotransferase (*p* = 0.0173) and alanine aminotransferase (*p* = 0.0378) levels and nausea (*p* = 0.0177) were more frequent in the elderly group than in the non-elderly group. However, in most cases, these AEs were transient and easily manageable. The most frequent grade 3 or 4 AE in both groups was neutropenia. A 67-year-old man with BR PC developed grade 5 pneumonitis. He had multiple comorbidities, including an old myocardial infarction, chronic obstructive pulmonary disease, and diabetes mellitus. After four treatment cycles, he presented with a high fever, worsening dyspnea, and cough. He died of interstitial and bacterial pneumonitis despite optimal medical management. 

### 3.4. Perioperative and Postoperative Outcomes

Perioperative and postoperative outcomes are summarized in Table 4. Surgical procedures, the median operation time, and blood loss were similar between the groups. The incidence of major postoperative complications (C-D > IIIA), including pancreatic fistula, did not differ between the two groups, resulting in a similar length of postoperative hospital stay. There was no death in either group. R0 resection was performed in 90% and 86% of elderly and non-elderly patients, respectively. The pathological T and N statuses did not differ between the two groups. Surgical outcomes of pancreaticoduodenectomy (PD) and distal pancreatectomy (DP) are summarized in Appendix A, respectively. There were no significant differences in surgical outcomes between elderly and non-elderly groups for PD and DP. The association between postoperative complications and G8 score is shown in Appendix A. There were no significant differences between the two groups.

### 3.5. Survival Outcomes

Figure 1A,B show the Kaplan–Meier curves obtained by the intention-to-treat-based analysis. The median PFS from the first day of NAC was 14.6 months (95% confidence interval (CI): 12.1-NA) in the elderly group versus 18.0 months (95% CI: 11.9–24.7) in the non-elderly group (*p* = 0.8260). OS was similar in the elderly and non-elderly groups (median OS: 27.5 months (95% CI: 18.3-NA) versus 27.6 months (95% CI: 23.2-NA), *p* = 0.6719). In the 49 patients who underwent pancreatectomy, the median RFS and OS in the elderly group were 11.1 (95% CI: 8.4-NA) and 23.9 months (95% CI: 19.8-NA), respectively, and the corresponding values in the non-elderly group were 16.3 months (95% CI: 14.1-NA) (*p* = 0.6647) and 32.0 months (95% CI: 23.8-NA), respectively (*p* = 0.2500; Figure 1C,D).

Table 5 shows the results of univariate and multivariate analyses of prognostic factors for overall survival. Multivariate Cox regression analysis revealed that ASA-PS and surgical resection are significantly associated with OS. Age > 75 years did not correlate with survival.

## 4. Discussion

In this study, we analyzed the outcomes in elderly patients with PC who were treated with neoadjuvant mFOLFIRINOX and compared them with those in non-elderly patients. The safety and feasibility of the treatment in elderly patients were similar to those in non-elderly patients. Therefore, age should not be a limiting factor for NAC application.

Compared with upfront surgery, NAC has been shown to improve survival outcomes in patients with R and BR PC [5,16,17,18,19,20]. However, there are few studies on the efficacy of chemotherapy including mFOLFIRINOX, which is an aggressive multi-agent chemotherapy, in elderly patients aged >75 years with PC. Additionally, the effectiveness of pancreatic surgery in this age group is also not well documented [17,21,22,23]. Oba et al. compared the outcomes in elderly patients with BR PC and those with locally advanced PC who received neoadjuvant FOLFIRINOX or GnP. The study population was divided into three age groups: <70, 70–74, and >75 years. They concluded that the safety and efficacy of NAC in elderly patients were similar to those in young patients. However, in their study, elderly patients aged >75 years were less likely to receive FOLFIRINOX compared with patients aged 70–74 and <70 years (35% vs. 50% vs. 80%, *p* < 0.001). No treatment-related FOLFIRINOX toxicity was observed. FOLFIRINOX may be selected for administration in elderly patients with a good physical status. In our study, an mFOLFIRINOX regimen was used considering the increased toxicity of the FOLFIRINOX regimen. The major grade 3 or 4 AE in our study was neutropenia (prevalence of 46% in the elderly group and 48% in the non-elderly group). These rates were similar to those reported in previous metastatic cohort studies [9]. Because severe neutropenia was manageable with granulocyte colony-stimulating factor, febrile neutropenia was observed in one patient in each group. Surgical resection following NAC was performed in 87% and 74% of the elderly and non-elderly patients, respectively. These results suggested that neoadjuvant mFOLFIRINOX was tolerable for patients with PC including those aged >75 years. A grade 5 pneumonitis was observed in one non-elderly patient with BR PC. NAC probably should not have been administered to this patient, as he had multiple comorbidities, including an old myocardial infarction, chronic obstructive pulmonary disease, and diabetes mellitus. Patients receiving NAC should be carefully selected considering their comorbidities.

Pancreatic surgery is one of the most challenging and complex types of abdominal surgery. Although improvements in surgical techniques and perioperative management have reduced hospital mortality, the rates of postoperative complications such as pancreatic fistula, bile leakage, and delayed gastric emptying remain high (up to 50%) [24,25,26]. Compared with non-elderly patients, elderly patients often have concomitant conditions and are considered to have higher risks associated with surgery. Some studies reported that the rates of complications and mortality in elderly patients were higher than the corresponding rates in younger patients, while others reported that they were comparable [24,27,28,29,30,31]. Our study showed no difference in the occurrence of POPF and major postoperative complications (C-D > IIIA) between the elderly and non-elderly groups. Furthermore, there was no postoperative mortality and the duration of postoperative hospital stay was similar between the two groups. Pancreatectomy following NAC can be safely performed, even in elderly patients. 

In the intention-to-treat-based analysis, PFS and OS rates in the elderly group treated with NAC were similar to those in the non-elderly group. Among the 49 patients who underwent pancreatectomy, no differences in either RFS or OS were observed between different age groups. Previous studies have reported poor OS in elderly patients who underwent pancreatectomy for PC, although the postoperative complication rates were similar between elderly and non-elderly patients [31,32]. The authors demonstrated that the rate of adjuvant chemotherapy (AC), a prognostic factor for PC, was significantly lower in elderly patients than in young patients. S-1 monotherapy, which is the standard AC in Japan, was used in our study [33]. Our study showed no difference in the rate of AC between the elderly and non-elderly groups. Therefore, elderly patients may have comparable prognoses in terms of OS and RFS. In Western countries, AC with mFOLFIRINOX has been considered the standard therapy based on the results of the PRODIGE-24 trial [34,35]. However, grade 3 and 4 AEs developed in 75.9% of patients who received mFOLFIRINOX. Therefore, this regimen should be limited to patients with good performance status. Indeed, some elderly patients may not complete AC because of postoperative complications, impaired organ function, or poor tolerance. In contrast, NAC offers some advantages such as (1) downstaging of the size of the primary tumors, (2) early administration and high tolerance of systemic therapy, (3) treatment of clinically undetectable microscopic metastatic disease, and (4) avoidance of unnecessary surgery in patients with progressive disease. In a recent large cohort study using the National Cancer Database, 5086 patients with PC aged >70 years were allocated to upfront surgery and AC (51.7%), upfront surgery only (29.9%), and NAC (18.4%) groups [36]. One-third of patients treated with NAC received AC. NAC was associated with a higher R0 resection rate and better OS than upfront surgery with or without AC. NAC may be an effective strategy for improving R0 resection and prognosis in elderly patients. 

This study had several limitations. First, this was a retrospective study conducted at a single institution, resulting in a limited number of patients and selection bias. The elderly patients in this study might have had a good general condition because 65% of the elderly patients had a 0–1 CCI and 78% had a G8 of ≤14. Therefore, it is necessary to conduct comprehensive geriatric assessments rather than performing age-based analyses alone. Prehabilitation programs should be considered according to physical and psychological status. Second, NAC dose reduction or discontinuation had no specific criteria. Third, the follow-up period was short, and long-term follow-up is needed to reveal clinically relevant survival differences. To overcome these limitations, further large-scale prospective studies are required.

## 5. Conclusions

The perioperative and postoperative outcomes of elderly patients who received neoadjuvant mFOLFIRINOX were comparable to those of non-elderly patients. This study suggests that NAC, including the mFOLFIRINOX regimen, should be considered a feasible treatment option even in elderly patients with PC.

## Figures and Tables

**Figure 1 cancers-16-02522-f001:**
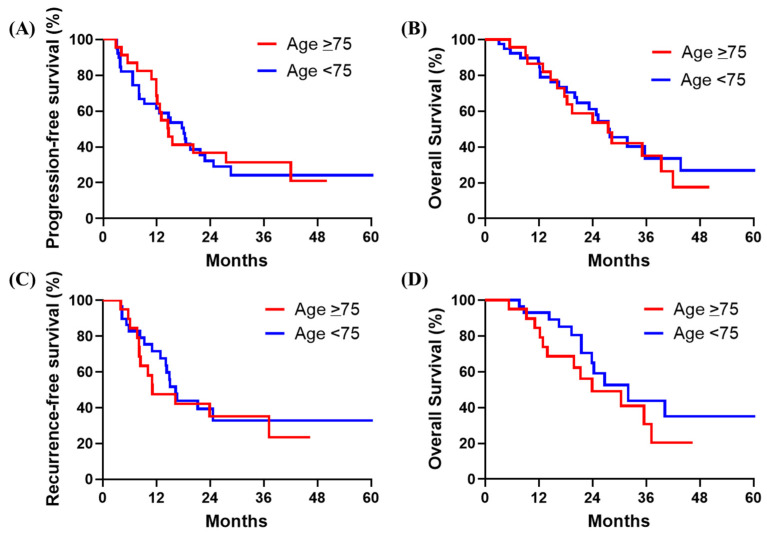
Kaplan–Meier curves by age group of patients treated with neoadjuvant chemotherapy (NAC): (**A**) Overall survival (OS) from the first day of NAC in the intention-to-treat analysis. (**B**) Progression-free survival (PFS) from the first day of NAC in the intention-to-treat analysis. (**C**) Recurrence-free survival (RFS) from the date of surgery among patients who underwent resection. (**D**) OS from the date of surgery among patients who underwent resection.

**Table 1 cancers-16-02522-t001:** Clinical characteristics before neoadjuvant chemotherapy.

Factors	Age ≥ 75 Years (n = 23)	Age < 75 Years (n = 39)	*p*-Value
Age (years)	79 (75–82)	67 (37–74)	<0.0001
Sex (Male)	12 (52)	23 (59)	0.7913
ASA-PS			0.2383
1	0	4 (10)	
2	18 (78)	25 (64)	
3	5 (22)	10 (26)	
BMI (kg/m^2^)	21 (16–26)	22 (15–32)	0.1101
CCI			0.9125
0	6 (26)	12 (31)	
1	9 (39)	15 (38)	
>2	8 (35)	12 (31)	
G8 score			0.3957
<14	18 (78)	26 (67)	
>14	5 (22)	13 (33)	
Tumor location			0.1164
Head	9 (39)	24 (62)	
Body/tail	14 (61)	15 (38)	
Resectability status			0.5082
R	20 (87)	30 (77)	
BR	3 (13)	9 (23)	
Initial tumor size (mm)	24 (13–44)	23 (11–57)	0.9509
Initial CA19-9 (U/mL)	81 (2–4701)	281 (2–19072)	0.4493
Initial SUV-max	5 (1.6–14.5)	5.5 (1.6–12.7)	0.9625
UGT1A1			0.7406
Wild	12 (52)	20 (51)	
6 hetero/28 hetero	11 (48)	18 (46)	
6 homo	0 (0)	1 (3)	

ASA-PS, American Society of Anesthesiologists physical status; BMI, body mass index; CCI, Charlson Comorbidity Index; R, resectable; BR, borderline resectable; CA19-9, carbohydrate antigen 19-9; SUV, standardized uptake value; UGT1A1, uridine diphosphate-glucuronosyltransferase 1A1.

**Table 2 cancers-16-02522-t002:** Outcomes of neoadjuvant chemotherapy.

	Age ≥ 75 Years (n = 23)	Age < 75 Years (n = 39)	*p*-Value
Completion of NAC	17 (74)	30 (77)	>0.9999
Dose reduction	21 (91)	28 (72)	0.1063
ARDI	65.3 (44.3–100.0)	77.0 (45.2–100.0)	0.0665
Surgical resection feasible	20 (87)	29 (74)	0.3381
Reason for not undergoing surgical resection			0.7008
Disease progression	2 (9)	7 (18)	
Poor general condition	0 (0)	1 (3)	
Adverse event	0 (0)	1 (3)	
Others	1 (4)	1 (3)	

NAC, Neoadjuvant chemotherapy; ARDI, average relative dose intensity.

**Table 3 cancers-16-02522-t003:** Adverse events.

	Age ≥ 75 Years (n = 23)		Age < 75 Years (n = 39)		*p*-Value	
	Any Grade	Grade ≥ 3	Any Grade	Grade ≥ 3	Any Grade	Grade ≥ 3
Neutropenia	18 (78)	11 (48)	26 (67)	18 (46)	0.3957	>0.9999
Anemia	21 (91)	0 (0)	32 (82)	2 (5)	0.4639	0.5256
Thrombocytopenia	16 (70)	0 (0)	28 (72)	2 (5)	>0.9999	0.5256
Febrile neutropenia	1 (4)	1 (4)	1 (3)	1 (3)	>0.9999	>0.9999
AST increased	5 (22)	0 (0)	21 (54)	2 (5)	0.0173	0.5256
ALT increased	7 (30)	0 (0)	23 (59)	2 (5)	0.0378	0.5256
Creatinine increased	2 (9)	0 ( 0)	2 (5)	2 (5)	0.6232	0.5256
Fatigue	10 (43)	2 (9)	23 (59)	1 (3)	0.2962	0.5494
Nausea	6 (26)	0 (0)	23 (59)	2 (5)	0.0177	0.5256
Biliary tract infection	0 (0)	0 (0)	4 (10)	4 (10)	0.2871	0.2871
Diarrhea	8 (35)	2 (9)	13 (33)	2 (5)	>0.9999	0.6232
Stomatitis	3 (13)	0 (0)	7 (18)	0 (0)	0.7313	>0.9999
Sensory neuropathy	8 (35)	0 (0)	21 (54)	0 (0)	0.1910	>0.9999
Pneumonitis	0 (0)	0 (0)	1 (3)	1 (3)	>0.9999	>0.9999

AST, Aspartate aminotransferase; ALT, alanine aminotransferase.

**Table 4 cancers-16-02522-t004:** Surgical and pathological outcomes.

	Age ≥ 75 Years (n = 20)	Age < 75 Years (n = 29)	*p*-Value
Operative procedure			0.4981
PD	9 (45)	18 (62)	
DP	10 (50)	10 (34)	
TP	1 (5)	1 (3)	
Vascular resection	2 (10)	3 (10)	>0.9999
Operation time (min)	451 (280–850)	523 (273–805)	0.2447
Blood loss (mL)	340 (25–2800)	475 (10–4450)	0.3098
pT stage			0.7145
0	1 (5)	0 (0)	
1	11 (55)	19 (66)	
2	8 (40)	8 (28)	
3	0 (0)	2 (7)	
pN1	9 (45)	6 (21)	0.1142
R0 resection	18 (90)	25 (86)	>0.9999
Clavien–Dindo ≥ IIIA	7 (35)	16 (55)	0.2450
POPF of ≥grade B	3 (15)	6 (21)	0.7199
DGE of ≥grade B	1 (5)	2 (7)	>0.9999
Mortality	0 (0)	0 (0)	>0.9999
Postoperative hospital stay (days)	19 (10–79)	24 (10–70)	0.3496
Adjuvant chemotherapy	18 (90)	27 (93)	>0.9999

PD, pancreaticoduodenectomy; DP, distal pancreatectomy; TP, total pancreatectomy; POPF, Postoperative pancreatic fistula; DGE, delayed gastric emptying.

**Table 5 cancers-16-02522-t005:** Univariate and multivariate analyses of prognostic factors for overall survival.

	Univariate Analysis		*p*-Value	Multivariate Analysis		*p*-Value
	HR	95% CI		HR	95% CI	
Age (≥75 vs. <75 years)	1.155	0.593–2.248	0.6723	1.247	0.608–2.556	0.5475
Sex (male vs. female)	1.307	0.658–2.593	0.4443			
ASA-PS (3 vs. 1/2)	1.678	0.803–3.508	0.1687	3.515	1.422–8.691	0.0065
BMI (<22 vs. ≥22)	1.182	0.613–2.277	0.6178			
CCI (≥2 vs. <2)	1.620	0.831–3.160	0.1566			
G8 (≤14 vs. >14)	1.586	0.722–3.482	0.2507			
Tumor location (Head vs. Body/tail)	1.404	0.721–2.733	0.3183			
Resectability status (BR vs. R)	2.663	1.198–5.923	0.0163			
Initial tumor size (≥20 vs. <20 mm)	1.442	0.677–3.073	0.3430			
Initial CA19-9 (≥37 vs. <37 U/mL)	1.855	0.843–4.085	0.1247	2.302	0.936–5.661	0.0695
Initial SUV-max (≥5 vs. <5)	1.709	0.880–3.319	0.1137			
Completion of NAC (no vs. yes)	1.918	0.906–4.062	0.0887			
Dose reduction (no vs. yes)	1.044	0.456–2.388	0.9195			
ARDI (≤70 vs. >70)	1.585	0.818–3.073	0.1725			
Surgical resection feasible (no vs. yes)	14.603	5.910–36.089	<0.0001	26.878	7.955–90.819	<0.0001

HR, hazard ratio; CI, confidence interval; ASA-PS, American Society of Anesthesiologists physical status; BMI, body mass index; CCI, Charlson Comorbidity Index; BR, borderline resectable; R, resectable; CA19-9, carbohydrate antigen 19-9; SUV, standardized uptake value; NAC, neoadjuvant chemotherapy; ARDI. Average relative dose intensity.

## Data Availability

The datasets used and analyzed during the current study are available from the corresponding author upon reasonable request.

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
