# Peer review of "Safety and Feasibility of Neoadjuvant-Modified FOLFIRINOX in Elderly Patients with Pancreatic Cancer"

_cancers, 2024, doi:10.3390/cancers16142522_

Round 1
Reviewer 1 Report
Comments and Suggestions for Authors
Authors of the article investigated mFOLFORINOX in elderly PC patients. The article investigates a very important question, that might introduce great interest. The following questions were raised:
Major:
1. Methods need some additions:
1A. Line 88-89: The presented grouping was not mentioned earlier, please give further information, why this was important. Later on, it was never used again.
1B. It was never mentioned, how AEs were defined. Did Authors use the CTCAE grading definitions? If yes, which version was used, etc.
1C. Similarly, Calvien Dindo was never defined in Methods.
2. Table 2: p-values are missing for the last parameters
3. Table 3: Please give details, what the p-values represent. E.g., did Authors conduct the comparison for neutropenia with this 2x2 table: 7-11 and 8-18?
4. Survival comparisons: please investigate the survival of patients not only with the logrank test, but with Cox regression as well. It can give further details, how the risk between the groups change with the time. Furthermore, advanced Cox modeling could be also required, e.g. the use of step functions. Moreover, it is strongly advised to create an extended Cox model, where the baseline hazards of the patiens are adjusted, e.g. for ARDI groups (such as: <50, 50-75, 75-90, 90-100). The latter can be achieved by using e.g. R with this code: summary(coxph(Surv(time, censor) ~ elderly_groups + strata(ARDI_groups), data = data)).
5. Did Authors performed any multivariate analysis?
6. Authors mentioned a case, where a patient could not be operated due to an AE. Please give further details both in results and discussion. What was this AE, was the operation feasable later, etc.
Minor:
1. Line 72: BRPC -> BR PC
2. There are a few expression, which could be abbreviated throughout the text, such as ASA-PS and BMI. Please, look through the manuscript, and abbreviate those terms.
3. In all of tables: Where a factor with multiple levels are presented, it is advised as per the MDPI author guidelines, that in the first column the in-between lines can be removed, and the cells for p-value can be merged. This way it is easier for the readers to read the tables.
4. Please provide the 95% confidence intervals fo the medain survivals.
5. Figure 1: in the figure captions, the text of 1C and 1D has been swapped
Comments on the Quality of English LanguageThere a few complex sentences that are a bit harder to read. Those have to be revised, if possible, split them into 2+ sentences. E.g. lines 182-185.
Line 79: PC NAC was initianted -> PC, and NAC was initated.
Table 2: Surgical resection -> surgical resection feasable
Line 223: which is standard -> which is the standard
Line 227: result of PRODIGE-24 -> result of the PRODIGE-24
Reviewer 2 Report
Comments and Suggestions for Authors
Overall this is a very interesting and relevant manuscript with regards to experience of modified Folfirinox in elderly patients. The study coming from Japan even though it doesnt have any novel findings verifies and adds to the existing literature. I do not see any major issues with regards to the manuscript. I recommend some minor editing for example in table 1 giving data only for one of the two genders is enough as this variable has only two values. Same thing for table 2 please give the number only for the resected cases as the rest are implied to be the non resected.
Since the authors have info on the G8 data for these patients can you see if the results of the G8 correlated with postop complications such as delirium etc irrespectively from the age group of the patient? that would add some novelty to the paper
In lines 46-47 please comment/cite that pancreas cancer is projected to be the second leading cause from cancer mortality in the west
Comments on the Quality of English Languageacceptable
Reviewer 3 Report
Comments and Suggestions for Authors
A good retrospective review of 23 patients 75 and older receiving mFOLFIRINOX for resectable and borderline resectable pancreatic cancer.
1. Add G8 as appendix or hyperlink
2. Expand on disease progression, especially in younger patients. Where and when did it occur. Why the trend (p=.066) in younger patients?
2. Shorter op times and less blood loss in older patients - does this reflect relatively more left pancreatectomies in that group? What happens when you compare whipple only in young vs. old patients? if trend continues, is there a bias in older patient towards smaller, more resectable tumors?
3. Define good physical status for older patients - G8<15? How would you approach borderline physical status? Do you have a prehabilitation program or philosophy?
A nice paper that should be published. Those of us in the West need to better understand how to extrapolate your experience to our largely more obese, more co-morbid population.
Round 2
Reviewer 1 Report
Comments and Suggestions for Authors
The manuscript improved significantly. I recommend its acceptance.